# Uncovering the reaction mechanism behind CoO as active phase for $CO_2$ hydrogenation

Iris C. ten Have [1], Josepha J. G. Kromwijk[1], Matteo Monai [1], Davide Ferri [2], Ellen B. Sterk[1], Florian Meirer [1✉] & Bert M. Weckhuysen [1✉]

Transforming carbon dioxide into valuable chemicals and fuels, is a promising tool for environmental and industrial purposes. Here, we present catalysts comprising of cobalt (oxide) nanoparticles stabilized on various support oxides for hydrocarbon production from carbon dioxide. We demonstrate that the activity and selectivity can be tuned by selection of the support oxide and cobalt oxidation state. Modulated excitation (ME) diffuse reflectance infrared Fourier transform spectroscopy (DRIFTS) reveals that cobalt oxide catalysts follows the hydrogen-assisted pathway, whereas metallic cobalt catalysts mainly follows the direct dissociation pathway. Contrary to the commonly considered metallic active phase of cobalt-based catalysts, cobalt oxide on titania support is the most active catalyst in this study and produces 11% $C_{2+}$ hydrocarbons. The $C_{2+}$ selectivity increases to 39% (yielding 104 mmol $h^{-1}$ $g_{cat}^{-1}$ $C_{2+}$ hydrocarbons) upon co-feeding CO and $CO_2$ at a ratio of 1:2 at 250 °C and 20 bar, thus outperforming the majority of typical cobalt-based catalysts.

[1] Inorganic Chemistry and Catalysis Group, Debye Institute for Nanomaterials Science, Utrecht University, Universiteitsweg 99, 3584 CG Utrecht, The Netherlands. [2] Paul Scherrer Institute, Forschungsstrasse 111, 5232 Villigen, Switzerland. ✉email: f.meirer@uu.nl; b.m.weckhuysen@uu.nl

With rising $CO_2$ levels in the atmosphere leading to climate change, it is of high interest to investigate methods to reduce the amount of anthropogenically emitted $CO_2$. A transition to a greener energy mix and to more sustainable processes for chemical production is on the way, but it will require years or perhaps even decades and huge investments to permeate the market. Moreover, some sectors intrinsically emit $CO_2$ (e.g., cement industry). Carbon capture and storage (CCS) and carbon capture and utilization (CCU) can be used to help curb persisting $CO_2$ emissions[1–3]. CCS is an efficient strategy to cut $CO_2$ emissions and store carbon in geological formations, but this technology is energy intensive and expensive[4]. Therefore, CCU is a more attractive and promising option[1]. Captured $CO_2$ can be used as a renewable resource to produce e.g., long-chain hydrocarbons, which can be used as transportation fuels. However, due to the thermodynamic stability of $CO_2$ the use of this greenhouse gas as a chemical feedstock is currently limited to a small number of industrial processes. For instance, the synthesis of urea and its derivatives, salicylic acid and carbonates[5], as well as more recently to the synthesis of methane in Power-to-Methane plants[6]. Another example is the methanol synthesis process, in which $CO/CO_2/H_2$ mixtures are converted to methanol with a $Cu/ZnO/Al_2O_3$ catalyst[7,8]. Several approaches like the photochemical[9], electrochemical[10–12], and thermochemical[13–17] conversion of $CO_2$ into more valuable long-chain hydrocarbons have been investigated. However, up to now only thermochemical $CO_2$ conversion has been proven to produce hydrocarbons longer than methane at high conversion yields[14,15], although Cu is known to electrochemically produce ethanol and ethylene[10,18,19]. Several catalysts, mainly based on Ni, Fe, Ru, Rh, Pt, and Pd, have been investigated, but most of them produce mainly methane, as is the case for Ni[20–23]. Creating products with longer hydrocarbon chains than methane is beneficial because they store more energy[24] and are easier to transport off-grid compared to gaseous methane. Creating long-chain hydrocarbons from $CO_2$ is thus a promising pathway toward a circular economy and will be useful in the next decade to produce, for example, fuels for aviation and for diesel engines.

Cobalt is an interesting candidate to investigate, since it has high C–C coupling activity in the similar CO hydrogenation reaction. This industrial process, also known as the Fischer–Tropsch synthesis (FTS), converts CO and $H_2$, better known as syngas, into e.g., fuels and chemicals with iron- or cobalt-based heterogeneous catalysts[25]. Iron generally produces lower olefins and oxygenates, whereas cobalt produces mainly long-chain paraffins. There are several parameters that influence the performance of FTS catalyst materials. For the cobalt-based FTS, the optimum cobalt nanoparticle size has been reported to be between 6 and 10 nm[26–28]. For nanoparticles smaller than 6 nm, the activity is generally lower and the selectivity toward $CH_4$, an unwanted product in the FTS, is higher. For catalysts with cobalt nanoparticles larger than 10 nm, the turnover frequencies (TOFs) were comparable to catalysts with 6–10 nm particles. For $CO_2$ hydrogenation, it has been reported that 10 nm cobalt particles display higher TOFs compared to 3 and 7 nm particles[29]. Besides nanoparticle size, the cobalt oxidation state, the cobalt phase, and the support oxide used to stabilize the metal nanoparticles greatly influence the activity and selectivity of the resulting catalyst[30,31]. $CoO/TiO_2$ has for example been reported to be more active in CO and $CO_2$ hydrogenation compared to its metallic equivalent[32–34]. However, $Co/SiO_2$ was found to be more active with metallic cobalt[32]. In another study, it was found that $Co/Al_2O_3$ converted $CO_2$ into ethanol with high selectivity due to coexisting Co and CoO phases[35]. Moreover, a well-balanced coexistence of Co and CoO on $SiO_2$ support, with cobalt phyllosilicate structure, has also been reported to exhibit high methanol selectivity in the $CO_2$ hydrogenation reaction[36].

Recently, Parastaev et al. were able to improve the $CO_2$ methanation activity of $Co/CeO_2$ by tuning the calcination temperature to create optimal metal-support interactions[37].

Elucidating the reaction mechanisms for hydrocarbon production from $CO_2$ remains elusive due to the complexity of the process and the large number of species involved[2,38,39]. For FTS catalysts, such as Co, a two-step process has been proposed: $CO_2$ is first converted to CO via the reverse water-gas-shift (RWGS) and then transformed into hydrocarbons through FTS[39]. Several mechanisms have been suggested for the RWGS and FTS individually, but most likely it is the nature of the catalyst that determines which of the pathways is dominant. For the RWGS, the direct dissociation pathway (also known as the redox or carbide mechanism) and the hydrogen(H)-assisted pathway (also known as the associative or formate mechanism) have been proposed[2,38]. The direct dissociation is facilitated by adsorbed CO as intermediate, whereas the H-assisted pathway is enabled by carbonate, formate, and formyl intermediates[2,38]. These intermediates can either be fully hydrogenated to form methane or they can undergo chain propagation via the FTS to form long-chain hydrocarbons, like paraffins and olefins. To investigate this, infrared (IR) spectroscopy is a promising tool for mechanistic studies[2,40], as it can probe the molecular vibrations of surface intermediates and active species. Nevertheless, mechanistic investigations remain challenging owing to sensitivity limitations of analytical tools, especially under relevant reaction conditions.

In this study, the influence of both reducible ($TiO_2$ and $CeO_2$) and non-reducible ($SiO_2$ and $Al_2O_3$) metal oxide supports and the effect of the cobalt oxidation state (CoO versus metallic Co) were investigated. Catalytic tests showed that metallic Co was typically more active than CoO, except for $Co/TiO_2$. Besides, CoO possessed less hydrogenation ability than metallic Co, resulting in the formation of $C_{2+}$ olefins rather than $C_{2+}$ paraffins. Using operando modulated excitation (ME) diffuse reflectance infrared Fourier transform spectroscopy (DRIFTS) with phase-sensitive detection (PSD) we observed that CoO catalysts followed the H-assisted pathway, whereas metallic Co catalysts followed the direct dissociation pathway. $Co/TiO_2$ was the most active catalyst in both oxidized and reduced state. For reduced $Co/TiO_2$, this was explained based on the red shift of the $CO_{ads}$ peak by 14 $cm^{-1}$, indicating a weaker C–O bond when changing from $Co/SiO_2$ to $Co/TiO_2$. In the case of the most active catalyst, $CoO/TiO_2$, the $C_{2+}$ selectivity could be improved from 11 to 39% upon co-feeding $CO:CO_2$ at a ratio of 1:2—this lead to a high overall $C_{2+}$ yield of 104 mmol $h^{-1}$ $g_{cat}^{-1}$ at 17.5% carbon conversion, $T = 250\,°C$, $P = 20$ bar, and a gas hourly space velocity (GHSV) of 3000 $h^{-1}$.

## Results

**Dependence of performance on cobalt oxidation state and support.** To investigate the dependence of activity and selectivity on the type of support oxide in the cobalt-based $CO_2$ hydrogenation reaction, we compared cobalt nanoparticles supported on $SiO_2$, $Al_2O_3$, $TiO_2$, and $CeO_2$. Physico-chemical properties of the support materials, such as surface area and pore size, can be found in Supplementary Table 1. To avoid interfering particle size effects on the activity, we ensured that the average cobalt particle size was above 10 nm in all catalysts. The $SiO_2$, $Al_2O_3$, and $TiO_2$-supported catalysts contained cobalt particles of similar sizes (14–17 nm), whereas the $CeO_2$-supported cobalt particles were larger (37 nm) (Fig. 1a–h). All catalysts contained $Co_3O_4$ after calcination, as determined with X-ray diffraction (XRD) (Supplementary Fig. 4 and Supplementary Table 2) and Raman microspectroscopy (Supplementary Figs. 5–8 and Supplementary Table 3). Catalytic testing in a fixed bed reactor at $T = 250\,°C$ and $P = 20$ bar was conducted using CoO (suffix: -ox) and metallic Co

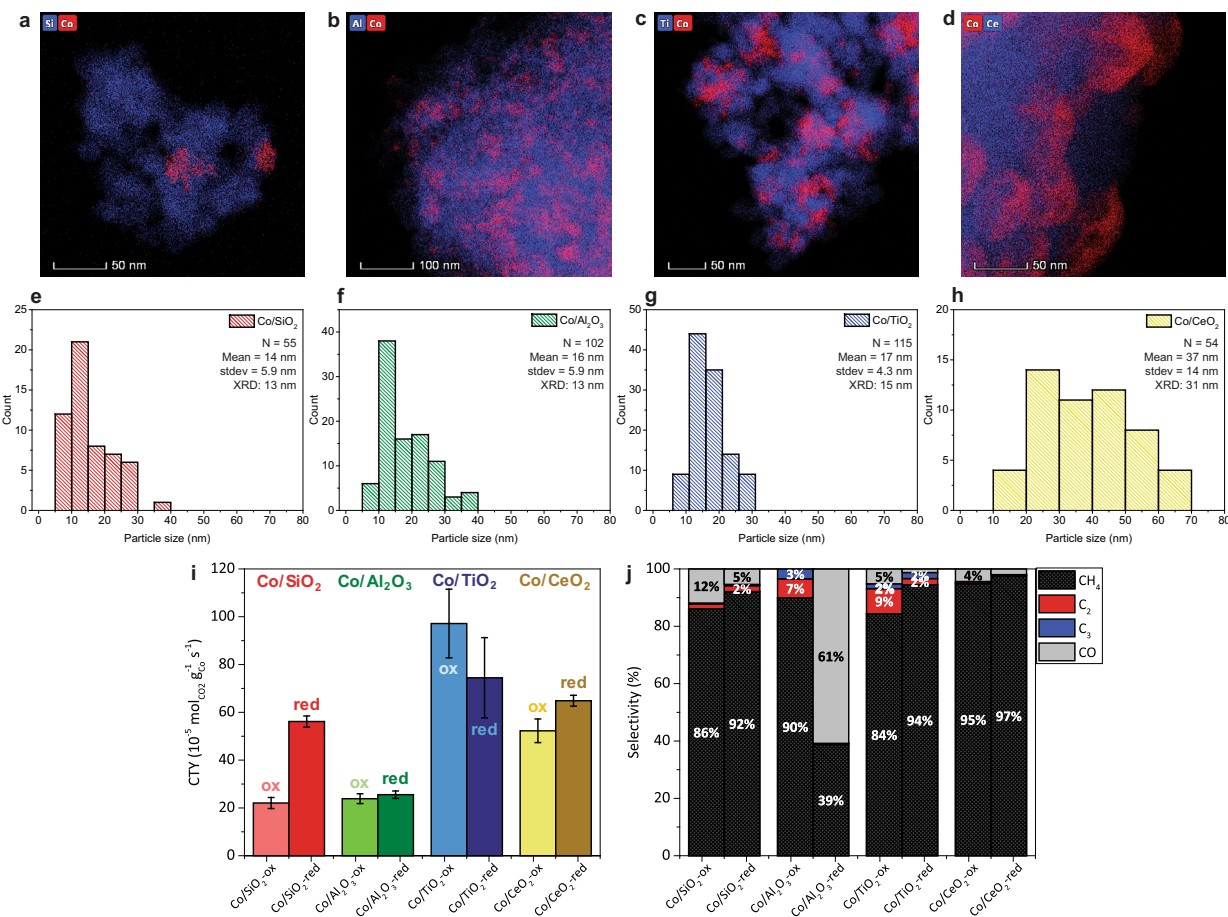

**Fig. 1 Cobalt particle size analysis and effect of support and oxidation state on catalytic activity.** Energy dispersive X-ray spectroscopy (EDX) elemental mapping of the calcined catalysts (250 °C in $N_2$): **a** Co/SiO$_2$, **b** Co/Al$_2$O$_3$, **c** Co/TiO$_2$, and **d** Co/CeO$_2$. Cobalt oxide particle size distributions determined with transmission electron microscopy (TEM), where $N$ denotes the number of particles counted, and average crystallite sizes calculated based on X-ray diffraction (XRD) results for **e** Co/SiO$_2$, **f** Co/Al$_2$O$_3$, **g** Co/TiO$_2$, and **h** Co/CeO$_2$. **i**, **j** Catalytic activity (cobalt-time-yield (CTY)) and selectivity of cobalt-based catalysts as measured with gas chromatography (GC) over 10 h time-on-stream in CoO (suffix: -ox) and metallic Co (suffix: -red) state at $T = 250$ °C, $P = 20$ bar, $H_2/CO_2 = 3$. The automated GC injections were performed every 23 min and the whiskers indicate the standard deviation (stdev) in the 26 data points per sample.

(suffix: -red) catalysts. Prior to testing, the indicated cobalt oxidation states were obtained by heating in 33 vol% $H_2/N_2$ at 250 °C for the CoO and 450 °C for the metallic Co catalysts. The oxidation state was verified and monitored with operando Raman micro-spectroscopy experiments (Supplementary Figs. 5 and 6). Additionally, $H_2$-temperature programmed reduction (TPR) was performed to assess the reducibility of all catalysts (Supplementary Fig. 9) and $CO_2$-temperature programmed desorption (TPD) was used to determine the basicity of the support materials (Supplementary Fig. 10 and Supplementary Table 4).

All catalysts displayed higher cobalt-time-yield (CTY) in their metallic state, except for Co/TiO$_2$, which was more active in its CoO state (Fig. 1i). The CeO$_2$-supported catalysts displayed a high selectivity toward methane (95–97%) in both metallic Co and CoO state. The SiO$_2$-supported catalyst had slightly lower methane selectivities of 92% and 86% in metallic Co and CoO state, respectively. The other products were CO, as well as C$_2$ and C$_3$ hydrocarbons. Co/Al$_2$O$_3$ mainly produced CO (61%) in metallic state and 10% C$_{2+}$ hydrocarbons in CoO state. Co/TiO$_2$ displayed the highest CTY in both metallic Co and CoO state. The Co/TiO$_2$-ox catalyst was the most active catalyst in this study with 11% selectivity to C$_{2+}$ hydrocarbons (see Supplementary Table 5 for more detailed information and additional standard deviations for the catalytic performance). The active phase in the

cobalt-based FTS has been debated for almost a century. Classically, metallic cobalt is believed to be the active phase in the FTS process[25]. However, our results showed that CoO on the reducible TiO$_2$ support is more active than metallic Co. This is in accordance with a study by Melaet et al. [32], where CoO/TiO$_2$ was more active than metallic Co for both CO and CO$_2$ hydrogenation. They attributed the phenomena to an interface formed between CoO and TiO$_2$ and they mentioned that strong metal-support interactions, where TiO$_x$ species encapsulate some of the active sites, could have decreased the activity of the metallic Co/TiO$_2$ catalyst. The reaction mechanisms at play, and especially whether or not these are different for CoO and metallic Co, are yet to be elucidated. In the section below, we will uncover active species and reaction mechanisms for the set of cobalt-based catalysts under study by means of operando ME DRIFTS with PSD.

**Influence of cobalt oxidation state and support on reaction mechanism.** Detecting active species with IR spectroscopy is often limited by the sensitivity of the characterization technique under reaction conditions and is generally challenging. Relevant signals could for example overlap strongly with irrelevant background signals, complicating the interpretation. To overcome the sensitivity problem, we used operando ME DRIFTS with PSD.

This method is based on the periodic variation of an external stimulus, in our case $CO_2$ gas, while IR spectroscopic data are recorded. The obtained spectra contain mixed signals of the active species, spectator species, deactivating species, and noise. The time-resolved IR data can be converted from the time domain to the phase domain by applying a set of mathematical transformations based on Fourier series, also known as PSD (Supplementary Fig. 3)[41,42]. By demodulating the periodically varying IR signals, the dynamic signals can then be separated from the static ones[43,44]. The spectator species and the noise are canceled out in the phase domain, as they do not exhibit a periodic response to the external stimulus. The resulting high-quality phase-resolved IR data only contain the periodically responding species and provide direct insights into the $CO_2$ hydrogenation mechanism, the nature of the active site(s), and kinetics.

The demodulated IR data can be found in Supplementary Fig. 12 and the corresponding mass spectrometry (MS) signals for $CH_4$ and $C_{2+}$ hydrocarbons obtained during the operando ME DRIFTS experiments can be found in Supplementary Fig. 13. Detailed peak assignments can be found in Supplementary Tables 7 and 8. In the averaged time-resolved IR spectra (Fig. 2a), (bi)carbonates, formates, and adsorbed CO (Fig. 2c–g) could already be observed on the cobalt-based catalysts, but the corresponding peaks were broad and convoluted. In general, the averaged time-resolved IR spectra of the catalysts with Co(O) supported on reducible supports displayed evident and broad signals of surface (bi)carbonates and formates, whereas the spectra of catalysts with Co(O) on non-reducible supports did not. This can be explained by the basicity of the supports, which we defined as the amount of $CO_2$ adsorbed per surface area unit measured with $CO_2$-TPD (Supplementary Fig. 10 and Supplementary Table 4). The basicity of the supports used in this study increases in the order $SiO_2 < Al_2O_3 < CeO_2 < TiO_2$. Interaction between basic $O^{2-}$ surface ions and $CO_2$ facilitates the formation of carbonates, whereas -OH surface groups enable the formation of bicarbonates from $CO_2$[46]. Surface vacancies, as present in large numbers on reducible supports like $TiO_2$ and $CeO_2$, aid the generation of formate species[45]. Besides, hydrogen spillover, replenishing e.g., –OH surface groups, is significant onto reducible supports, such as titania and ceria[47]. Additionally, weaker signals of adsorbed CO appeared around 2000 cm$^{-1}$ [48] in the averaged time-resolved IR spectra (Fig. 2a).

PSD revealed species on the cobalt-based catalysts that would otherwise not have been visible so clearly. This can be seen by comparing the averaged time-resolved IR spectra (Fig. 2a) and the phase-resolved amplitude spectra (Fig. 2b), that are composed of the absolute maxima (in the phase domain) at every single wavenumber[49]. The surface (bi)carbonates observed on Co/TiO$_2$ and Co/CeO$_2$ and to some extent on Co/Al$_2$O$_3$ were dynamic species that varied with the external stimulus and thus became visible in the phase-resolved amplitude spectra (Fig. 2b). However, we cannot unambiguously conclude that they actively take part in the conversion of $CO_2$ to hydrocarbons, as the adsorption and desorption of these species could as well lead to the appearance of dynamic signals. Peak splitting was observed for the carbonates on Co/TiO$_2$. For Co/TiO$_2$-red $\nu_{asym}(CO_3^{2-})$ split into 1362 and 1378 cm$^{-1}$ and $\nu_{sym}(CO_3^{2-})$ split into 1562 and 1574 cm$^{-1}$. Such splits have been observed in earlier studies[50–52] and ascribed to different types of coordination and/or different adsorption centers[50], suggesting the formation of an interfacial area between Co and TiO$_2$ with different adsorption properties.

The most evident difference between the CoO (suffix: -ox) and metallic Co (suffix: -red) catalysts was the presence of adsorbed CO around 2000 cm$^{-1}$ (Fig. 2c) only on all the metallic Co catalysts. In the averaged time-resolved spectra, adsorbed CO was visible as a broad band on the metallic cobalt catalysts (Fig. 2a).

Though, for Co/CeO$_2$-red this band could barely be observed, possibly due to the intense contributions of (bi)carbonates and/or formates. In the phase-resolved amplitude spectra (Fig. 2b), on the other hand, the adsorbed CO signals became clear and sharp signals with peak maxima that varied per support material. Hence, irrespective of the support material, the metallic Co catalysts mainly followed the direct dissociation mechanism (Fig. 2h), as indicated by the presence of adsorbed CO as an intermediate. The energy of the peak of linearly adsorbed CO ($\nu(CO)$) is a measure of the C=O bond strength. When CO is coordinated to a metal atom, the metal d-orbitals donate electrons to the $\pi^*$ orbital of CO (Fig. 3a), formally decreasing the bond order and weakening the C=O bond. Thus, the vibrational frequency of adsorbed CO decreases and the resulting wavenumber in the IR spectrum shifts down. In the series of the metallic cobalt catalysts, Co/TiO$_2$ exhibited the lowest wavenumber for $\nu(CO)$ (Fig. 2b), indicating that the C=O bond was the weakest and providing a plausible explanation for the highest activity of Co/TiO$_2$. Besides, based on the position of the C=O stretching vibration from the phase-resolved amplitude spectra, there was an apparent optimum for the support reducibility, as determined from H$_2$-TPR data (Supplementary Fig. 9), around Co/TiO$_2$. This is depicted in Fig. 3b. On the other hand, the CoO-containing catalysts did not show any adsorbed CO species. Instead, different types of formyl, formate, and carbonate species (Fig. 2d–g) were observed on their surfaces in the phase-resolved amplitude spectra (Fig. 2b). The CoO catalysts thus followed a different mechanism than the metallic Co counterparts, namely the H-assisted mechanism (Fig. 2h). This observation is in accordance with theoretical work, where they found that $CO_{ads}$ adsorption is strong on metallic Co ($-1.99$ eV = $-192$ kJ/mol) and weak on CoO ($-0.33$ eV = $-32$ kJ/mol)[53]. Consequently, the $CO_{ads}$ vibration is observed on metallic Co, but not on CoO. Besides, the CoO catalysts produced more olefinic $C_2$ and $C_3$ products, while metallic Co primarily produced paraffins (Supplementary Table 5). For example, Co/TiO$_2$-ox, exhibited olefin/paraffin ratios for $C_2$ and $C_3$ of 0.4 and 1.7, respectively. Co/TiO$_2$-red, on the other hand, almost exclusively produced paraffins. These results suggest that the hydrogenation steps are less favorable and/or that the hydrogen availability is lower on CoO compared to a metallic Co surface, which may positively affect C–C coupling. To underline that CoO and metallic Co are simply different surfaces when it comes to $CO_2$ adsorption, i.e., altering the $CO_2$ hybridization, we performed density functional theory calculations. Geometry optimization of $CO_2$ on Co(110) and CoO(100), the most active facets, led to negative adsorption energies of $-63.2$ kJ/mol, and $-34.1$ kJ/mol, respectively. The O–C–O bond angle, indicative of bond activation, deformed more on Co(110) compared to CoO(100). Besides, the C–O bond length, another indicator of bond activation, elongated more on Co(110) compared to CoO(100). A more elaborate discussion on this can be found in Supplementary Fig. 11, Supplementary Table 6, and the accompanying text.

**Kinetic insights from PSD analysis.** From the phase shift $\varphi$ we can derive kinetic information about the cobalt-based systems[42,54]. The phase shift describes the difference between the external stimulus and the maximum intensity of a responding active species. We used $\varphi$ as a diagnostic tool to identify the responding signals and to distinguish between different (overlapping) contributions within one signal. For the identified signals, we looked at the intensity in the time domain. More specifically, we followed the intensity decrease of the signals in the first 10 s after the $CO_2$ was turned off during the modulated experiment. The steepness of the slope of desorption was then

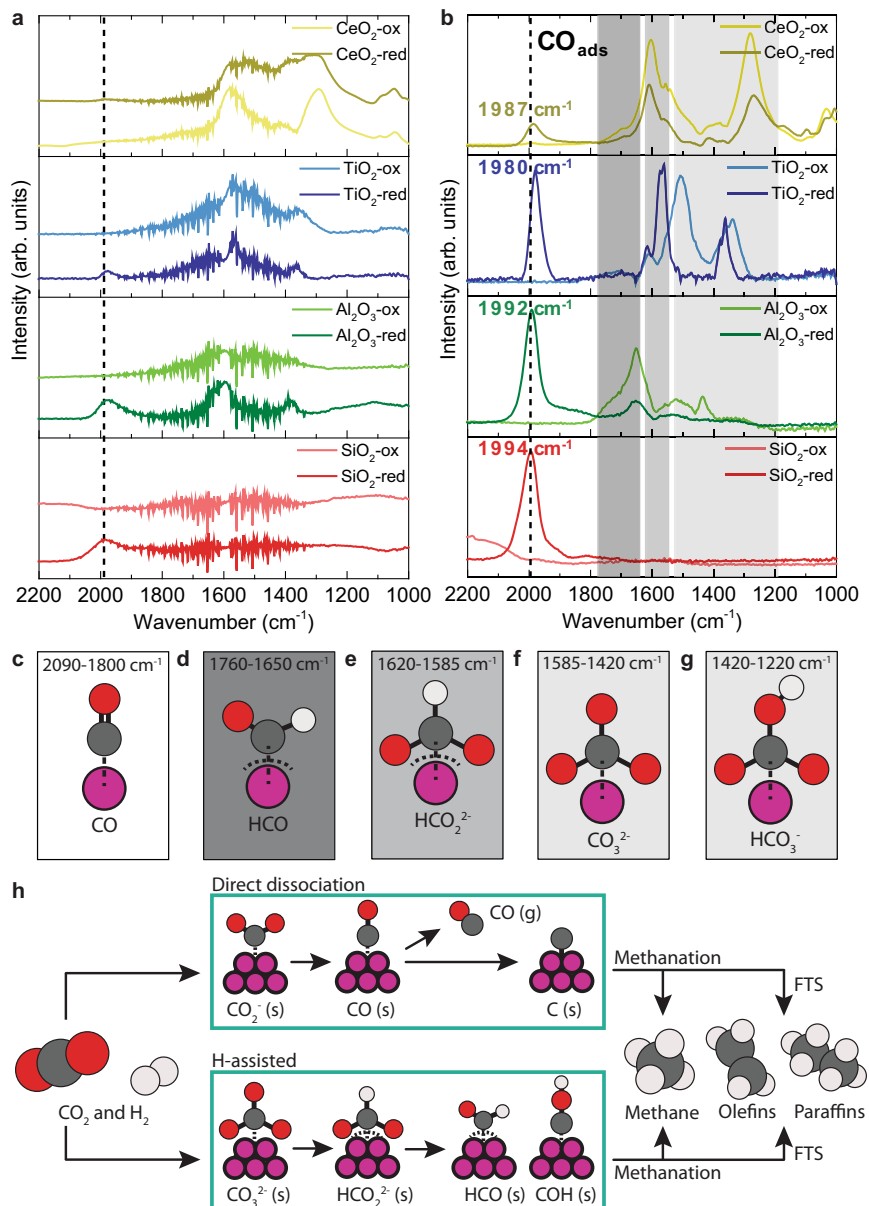

**Fig. 2 Modulated excitation diffuse reflectance infrared Fourier transform spectroscopy (ME DRIFTS) during CO$_2$ hydrogenation with the cobalt-based catalysts. a** Averaged time-resolved DRIFT spectra and **b** phase-resolved amplitude spectra of CoO (suffix: -ox) and metallic Co (suffix: -red) supported catalysts ($T = 250\,°C$, $P = 1\,bar$, $H_2/CO_2 = 3$). Adsorbed surface species with characteristic vibrational energies: **c** CO, **d** formyl, **e** formate, **f** carbonate, and **g** bicarbonate. **h** Simplified reaction pathways for cobalt-catalyzed CO$_2$ hydrogenation to hydrocarbons. In the direct dissociation mechanism CO$_{ads}$ is an intermediate, which either desorbs or forms C$_{ads}$. and then hydrocarbon products. The H-assisted mechanism involves surface carbonates, formates, and formyl as intermediates. The intermediates can either be fully hydrogenated to methane or converted into olefins or paraffins via C–C coupling (Fischer–Tropsch synthesis). Further details on experimental methodology and data analysis can be found in Supplementary Fig. 3.

used as a measure for relative kinetics; the steeper the slope, the faster the species. By comparing the responses of all the active species identified with PSD, we obtained relative kinetics of the species participating in the different CO$_2$ hydrogenation reaction mechanisms. We will explain the concept by focusing on Co/TiO$_2$, the most active catalyst in this study. The phase shifts and desorption slopes for all other catalysts can be found in Supplementary Figs. 12–17. We know from the phase-resolved amplitude spectra that Co/TiO$_2$-ox mainly followed the H-assisted pathway, while Co/TiO$_2$-red mainly followed the direct dissociation pathway (Fig. 4a, b). To visualize the relative kinetics of the different species, we plotted the desorption slopes of the respective signal intensity decrease during the first 10 s after

turning the CO$_2$ gas off for carbonate, formate, formyl, and adsorbed CO in Fig. 4c, d. Co/TiO$_2$-ox, as well as the other CoO-containing catalysts (Supplementary Figs. 12–17) displayed carbonate, formate (indicated in red in Fig. 4c, d), and formyl species in a similar kinetic regime. However, for Co/TiO$_2$-red, as well as for the other metallic Co catalysts, adsorbed CO (indicated in gray in Fig. 4d) displayed a steeper desorption slope, suggesting that CO responded faster than the carbonate, formate, and formyl species. This tells us that the direct dissociation pathway, of which adsorbed CO is a key intermediate, occurred at a higher rate than the H-assisted pathway. However, the product distributions (Fig. 1j) indicated that the H-assisted pathway was more beneficial for the production of C$_{2+}$ hydrocarbons. For example,

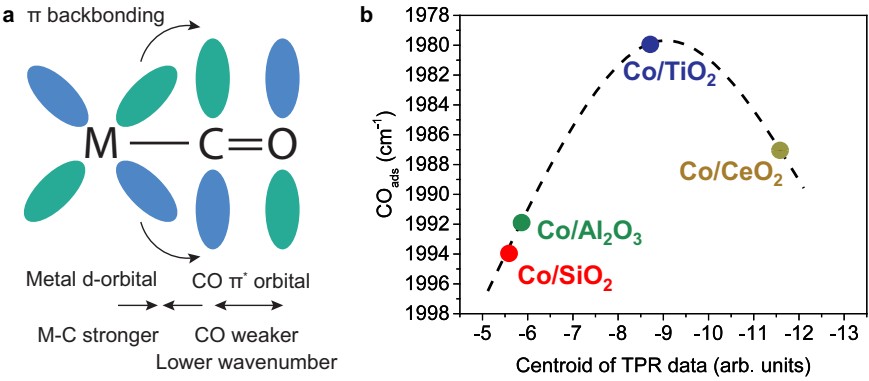

**Fig. 3 Support-dependent CO adsorption strengths for the metallic cobalt catalysts. a** Schematic drawing of π backbonding when C=O coordinates to a metal center. **b** Position of linearly adsorbed CO from the phase-resolved amplitude spectra versus the support reducibility, which was determined from $H_2$-temperature programmed reduction (TPR) data.

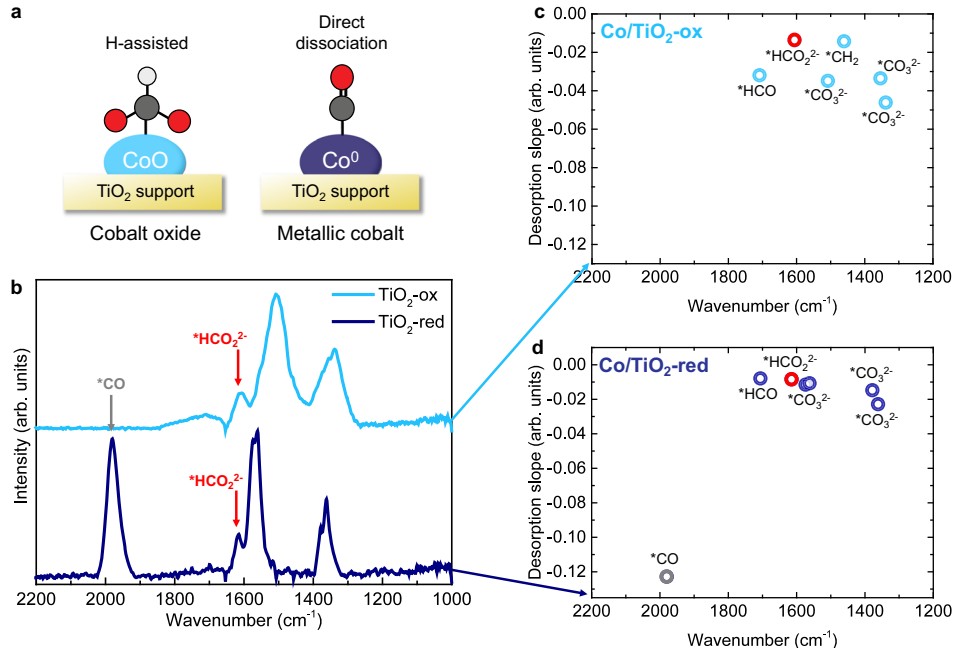

**Fig. 4 Kinetic information derived from phase-sensitive detection (PSD) analysis. a** Schematic representation of the H-assisted mechanism, dominant for CoO (suffix: -ox) catalysts, and the direct dissociation mechanism, dominant for metallic Co (suffix: -red) catalysts. **b** Phase-resolved amplitude spectra for Co/$TiO_2$-ox and Co/$TiO_2$-red. Desorption slopes (first 10 s after turning the $CO_2$ gas off) of selected species for **c** Co/$TiO_2$-ox and **d** Co/$TiO_2$-red. Both samples showed carbonate, formate (*$HCO_2^{2-}$ indicated in red; 1609–1615 cm$^{-1}$), and formyl species. Co/$TiO_2$-ox additionally showed *$CH_2$ species. For Co/$TiO_2$-red, *CO (gray; 1980 cm$^{-1}$) displayed faster kinetics (steeper slope) than the carbonate, formate, and formyl species. The direct dissociation mechanism was thus faster than the H-assisted mechanism.

Co/$Al_2O_3$-ox and Co/$TiO_2$-ox produced more $C_{2+}$ hydrocarbons compared to the metallic counterparts. Figure 5 depicts a schematic overview of elementary reaction steps in the $CO_2$ hydrogenation to methane and $C_{2+}$ hydrocarbons based on the intermediates detected on the Co/$TiO_2$ catalysts using ME DRIFTS.

**Understanding Co/$TiO_2$ via kinetic parameters**. For the best performing catalyst in our study, Co/$TiO_2$, we additionally determined a set of kinetic parameters at $P = 20$ bar in both the CoO and metallic Co state. The overall apparent activation energy ($E_a$) for $CO_2$ hydrogenation was slightly lower for Co/$TiO_2$-ox, 113 ± 3, compared to Co/$TiO_2$-red, 122 ± 5 (Table 1 and Supplementary Figs. 19 and 20). This is in line with the better performance of Co/$TiO_2$-ox compared to Co/$TiO_2$-red. Moreover, all the apparent activation energies for $CH_4$, $C_{2+}$, and CO were

considerably lower for Co/$TiO_2$-ox than for Co/$TiO_2$-red (Table 1). $C_{2+}$ products, for example, displayed an apparent activation energy of 98 ± 3 kJ/mol with the Co/$TiO_2$-ox catalyst, whereas Co/$TiO_2$-red resulted in a value of 115 ± 5 kJ/mol. To gain more insights, the reaction orders in $CO_2$ and in $H_2$ were determined for both samples. The higher reaction order in $CO_2$ of Co/$TiO_2$-ox (0.38 ± 0.09) compared to Co/$TiO_2$-red (0.15 ± 0.04) indicated that a strongly adsorbed intermediate derived from $CO_2$ on the Co/$TiO_2$-red surface, most likely adsorbed CO, hinders the reaction[55]. For metallic Co, a reaction order of 0.14 in $CO_2$ has been reported previously for the $CO_2$ hydrogenation reaction[56]. Besides, the reaction orders in $H_2$ were almost completely opposite for the 2 samples: a positive order of 1.24 ± 0.40 for Co/$TiO_2$-ox versus a negative order of −1.15 ± 0.07 for Co/$TiO_2$-red. This particularly substantiates the hypothesis that the Co/$TiO_2$-ox catalyst, following the H-assisted mechanism, benefits from a

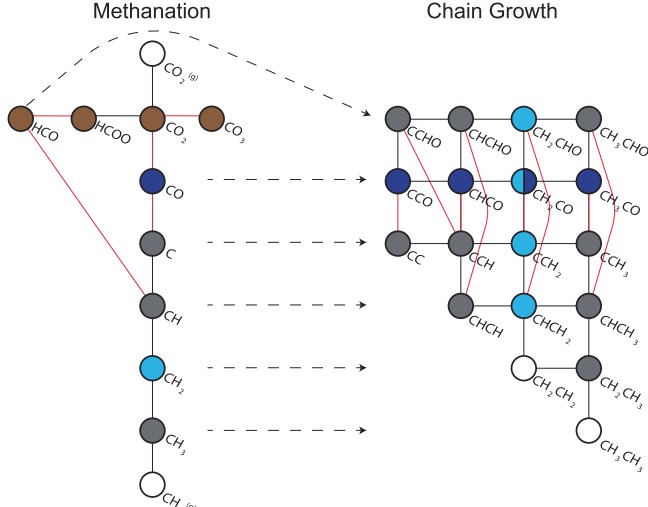

**Fig. 5 Schematic overview of elementary reaction steps in CO$_2$ hydrogenation on Co/TiO$_2$.** The white nodes indicate reactant and product molecules, the brown nodes correspond to reaction intermediates that were detected ME DRIFTS on both CoO and metallic Co. The light blue node for CH$_2$ on the left side was detected only on CoO and the dark blue node for CO only on metallic Co. On the right side the C–C coupled species with either CH$_2$ or CO are colored light blue or dark blue respectively. Black lines connecting the nodes represent (de)hydrogenation steps and the red lines indicate (de)coupling of an oxygen atom. The arrows with a dotted line represent C–C coupling events.

**Table 1 Kinetic parameters for Co/TiO$_2$.**

| Catalyst | | $E_a$ (kJ/mol)$^a$ | $R^2$ | CO$_2$ reaction order$^b$ | H$_2$ reaction order |
|---|---|---|---|---|---|
| Co/TiO$_2$-ox | Total | 113 ± 3 | 0.98 | 0.38 ± 0.09 | 1.24 ± 0.40 |
| | CH$_4$ | 113 ± 2 | 0.98 | 0.22 ± 0.11 | 1.36 ± 0.35 |
| | C$_{2+}$ | 98 ± 3 | 0.98 | 0.64 ± 0.07 | −0.13 ± 0.07 |
| | CO | 146 ± 7 | 1.00 | 0.49 ± 0.20 | 0.92 ± 0.13 |
| Co/TiO$_2$-red | Total | 122 ± 5 | 0.99 | 0.15 ± 0.04 | −1.15 ± 0.07 |
| | CH$_4$ | 121 ± 4 | 0.99 | 0.19 ± 0.05 | −1.21 ± 0.06 |
| | C$_{2+}$ | 115 ± 5 | 1.00 | 0.28 ± 0.16 | −1.13 ± 0.16 |
| | CO | 156 ± 8 | 0.99 | 1.04 ± 0.18 | −1.35 ± 0.20 |

$^a E_a$ was calculated with six data points at $T = 200$–$280\,°C$ and $P = 20$ bar. The $R^2$ values of the trendlines are reported in the third column.
$^b$CO$_2$ and H$_2$ reaction orders were determined at $T = 250\,°C$ and $P = 20$ bar.

higher partial pressure in H$_2$. On the other hand, the Co/TiO$_2$-red catalyst, following mainly the direct dissociation mechanism, benefits from a lower partial pressure in H$_2$, as H$_2$ may be competing with adsorbed CO, the most important intermediate in the direct dissociation mechanism.

**Shifting the product selectivity toward long-chain hydrocarbons.** With all these fundamental insights in hand, we are still left with the question how to directly obtain more long-chain hydrocarbon products from CO$_2$. In an attempt to answer this question, we took the best performing catalyst of this study, Co/TiO$_2$, and conducted experiments in which CO gas was co-fed at a CO$_2$ to CO ratio of 2. This approach was inspired by the industrial methanol synthesis process, where optimum performance of Cu/ZnO/Al$_2$O$_3$ catalysts is generally obtained by the synergistic effects of CO and CO$_2$ gas combined[7,8]. We found that upon co-feeding CO, the CO$_2$ conversion of Co/TiO$_2$-ox doubled and the C$_{2+}$ selectivity increased from 11 to 39%, leading to an

overall C$_{2+}$ yield of 104 mmol h$^{-1}$ g$_{cat}^{-1}$ (Fig. 6a and Supplementary Table 9). This catalyst even outperformed the majority of other cobalt-based catalysts that have been used for CO/CO$_2$ hydrogenation to C$_{2+}$ products (Table 2). The main fraction of the long-chain hydrocarbons was C$_2$ and C$_3$ (59%). However, C$_4$ (27%) and even C$_{5+}$ (14%) products were observed as well (Fig. 6a). For Co/TiO$_2$-red, the CO$_2$ conversion increased slightly more than for Co/TiO$_2$-ox upon co-feeding CO/CO$_2$, but the C$_{2+}$ selectivity only increased from 5 to 13%, leading to an overall C$_{2+}$ yield of 37.3 mmol h$^{-1}$ g$_{cat}^{-1}$. This suggests that the H-assisted mechanism is a more favorable pathway to produce long-chain hydrocarbons than the direct dissociation mechanism. This hypothesis was substantiated by comparing the olefin/paraffin ratios (Fig. 6b).

During CO$_2$ hydrogenation the Co/TiO$_2$-ox catalyst had a 0.4 and 1.7 olefin/paraffin ratio for C$_2$ and C$_3$ hydrocarbons, respectively. The Co/TiO$_2$-red catalyst, on the other hand, almost exclusively produced paraffins. This confirmed that CoO had a lower hydrogenation activity than metallic Co, explaining the higher selectivity to C$_{2+}$ products. This is in accordance with earlier observations that CoO possesses a lower hydrogenation activity compared to metallic Co and thus produced more olefins[32]. The concept is reminiscent of oxidic promoters, such as MnO, used in the FTS process to steer the selectivity from paraffins toward olefins[57–60]. Here, MnO decreases the cobalt reducibility, resulting in a more oxidic composition of the cobalt surface[59,60]. Such a surface favors β-hydrogen abstraction to produce olefins over α-hydrogen addition to produce paraffins[58,60,61]. And indeed, when Co/TiO$_2$-ox and Co/TiO$_2$-red were additionally tested under FTS conditions (Fig. 6), we found that the olefin/paraffin ratios for Co/TiO$_2$-ox were around 10 for C$_2$-C$_4$ products, whereas the ratios were only between 0.3 and 3.6 for Co/TiO$_2$-red. Co-feeding CO increased the olefin/paraffin ratios drastically for Co/TiO$_2$-ox and Co/TiO$_2$-red started producing some olefins as well (Fig. 6b). More details on the catalytic performance can be found in Supplementary Table 9 and the thermodynamic stability of the different cobalt phases under reaction conditions can be found in Supplementary Fig. 21. Co-feeding CO/CO$_2$ mixtures may thus be a profitable method to directly produce long-chain hydrocarbons from CO$_2$ at industrial scale. To assess the long-term stability of the Co/TiO$_2$-ox catalyst, it was tested at 250 °C and 20 bar for 150 h in total: first for 50 h under CO/CO$_2$ co-feeding conditions (CO$_2$/CO = 2) and then for 100 h under CO$_2$ hydrogenation conditions (H$_2$/CO$_2$ = 3) (Fig. 6c, d). For the 50 h of co-feeding, the total carbon conversion started at ~18% and stabilized after about 10 h to ~16%, while the C$_{2+}$ selectivity started at ~40% and stabilized at ~35%. For the following 100 h of CO$_2$ conversion only, the conversion started at ~7.0% and remained ~4.5% after 100 h, while the C$_{2+}$ selectivity increased from ~10% in the first few h to ~20% after 100 h, indicating that the activity loss over time was mostly related to a decrease in methane production. The long-term stability of Co/TiO$_2$-red over 150 h time-on-stream can be found in Supplementary Fig. 22. After 150 h time-on-stream, we verified with XRD that Co/TiO$_2$-ox contained CoO and Co/TiO$_2$-red contained metallic cobalt (face-centered cubic) (Supplementary Fig. 23).

## Discussion

We established that both metallic Co and CoO are active phases in the CO/CO$_2$ hydrogenation. To investigate the influence of the support oxide, we prepared a set of cobalt-based catalysts with both non-reducible supports (SiO$_2$ and Al$_2$O$_3$) and reducible supports (TiO$_2$ and CeO$_2$). We performed catalytic testing at industrially relevant conditions (T = 250 °C and P = 20 bar) and explained the

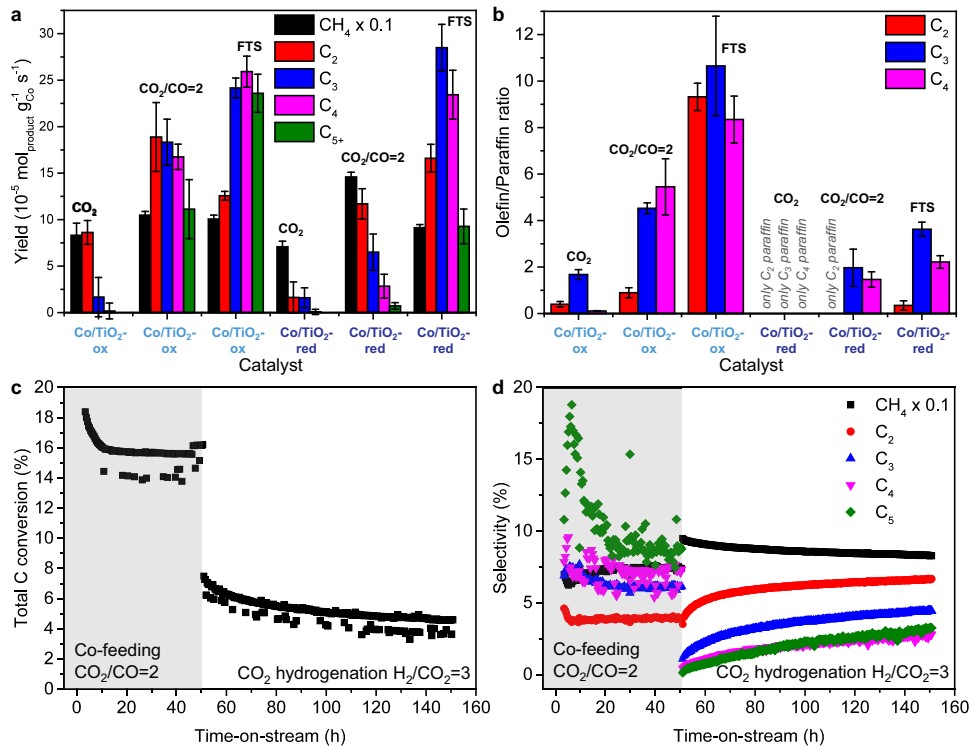

**Fig. 6 Co-feeding CO/CO₂ to Co/TiO₂ to shift the selectivity to long-chain hydrocarbons and long-term stability. a** Catalytic testing results ($T = 250$ °C, $P = 20$ bar, GHSV = 3000 h⁻¹, 10 h time-on-stream) comparing the product yields of $CO_2$ hydrogenation ($H_2/CO_2 = 3$) to $CO/CO_2$ ($H_2/CO_2/CO = 9/2/1$) hydrogenation and Fischer–Tropsch synthesis (FTS; $H_2/CO = 2$) for Co/TiO₂ containing CoO (suffix: -ox) or metallic Co (suffix: -red). **b** Olefin/Paraffin ratios for $C_2$, $C_3$, and $C_4$ products during $CO_2$ hydrogenation, $CO/CO_2$ ($H_2/CO_2/CO = 9/2/1$) hydrogenation, and FTS ($H_2/CO = 2$). The whiskers in **a** and **b** indicate the standard deviation. Stability test ($T = 250$ °C, $P = 20$ bar, GHSV = 3000 h⁻¹) of Co-TiO₂-ox operated for 150 h displaying **c** conversion and **d** selectivity first while co-feeding $CO/CO_2$ ($H_2/CO_2/CO = 9/2/1$) for 50 h and then during $CO_2$ hydrogenation only ($H_2/CO_2 = 3$) for 100 h.

observed phenomena by identifying active species using operando ME DRIFTS. The TiO₂ support provided the optimum reducibility for weakening C–O bonds and resulted in the highest $CO_2$ hydrogenation activity in this study. For most catalysts under study, metallic cobalt was more active than cobalt oxide, which is in accordance with the classical view of metallic cobalt as active phase. However, all catalysts with cobalt oxide were active as well. For Co/TiO₂, the catalyst containing CoO was even more active than metallic Co. We found that the catalysts with metallic Co mainly followed the direct dissociation pathway with adsorbed CO as a key intermediate. On the other hand, the catalysts with CoO mainly followed the H-assisted pathway via carbonate, formate, and formyl species. Although the direct dissociation was kinetically faster compared to the H-assisted pathway, the latter seemed more beneficial for the production of $C_{2+}$ hydrocarbons. The fundamental insights obtained here provide a paradigm shift in the classical view of the catalytically active phase. This has implications for the catalytic industry, as reductive pre-treatments at elevated temperature may not always be necessary. For the best catalyst in our study, CoO/TiO₂, we managed to double the $CO_2$ conversion and to shift the selectivity to $C_{2+}$ hydrocarbons from 11 to 39% by co-feeding CO and $CO_2$ at a ratio of 1:2. This led to a high overall $C_{2+}$ yield of 104 mmol h⁻¹ g$_{cat}$⁻¹ at $T = 250$ °C, $P = 20$ bar, and a GHSV of 3000 h⁻¹. This can thus be a promising way for industrial applications to directly produce long-chain hydrocarbons, instead of methane, from $CO_2$.

## Methods
**Catalyst synthesis**. Cobalt catalysts with a loading of ~10 wt% were prepared via incipient wetness impregnation. The aqueous metal precursor solution

(Co(NO₃)₂·6H₂O; Sigma-Aldrich, 99.999% trace metal basis), with a volume equal to the pore volume of the support, was added to the support material under vacuum and continuous stirring. The CeO₂ sample was prepared in two steps, as the solubility of Co(NO₃)₂·6H₂O was not sufficient to dissolve in a volume of water equal to the pore volume of the CeO₂ support. The resulting powder was dried in an oven at 60 °C overnight. Subsequently, the samples were calcined at 250 °C for 2 h (heating ramp of 5 °C min⁻¹) in a tube furnace in a N₂ flow of 100 ml min⁻¹. The support properties and exact amounts of chemicals used can be found in Supplementary Table 1.

**Transmission electron microscopy**. Transmission electron microscopy (TEM) was performed with a FEI Talos F200X. The TEM samples were prepared suspending the catalysts in absolute ethanol using sonication. Consequently, the suspension was dropcasted on a carbon/formvar-coated Cu grid (200 mesh). The microscope was operated at 200 kV and equipped with a high-brightness field emission gun (X-FEG) and a Super-X G2 energy dispersive X-ray (EDX) detector. The samples were analyzed with scanning (S)TEM combined with high-angle annular dark-field (STEM-HAADF). To determine the average cobalt particle size, the images were analyzed using the ImageJ software.

**X-ray diffraction**. XRD was performed using a Bruker D2 phaser equipped with a Co radiation source ($\lambda = 1.789$ Å). Diffraction patterns of calcined and spent catalysts were recorded between 5 and 85° 2Θ with an increment of 0.05° and 1 s/step. The average cobalt crystallite size was calculated from the peak at 43° using the Bruker EVA software.

**High-pressure catalytic testing**. Performance testing at 20 bar for 10 h was carried using in-house built high-pressure set-up. A steel reactor was filled with 200 mg of catalyst sample sieved to a grain size of 150–450 μm. The sample was plugged between two quartz wool plugs. The reactor was placed in an oven and connected to the gas inlet and outlet. A back-pressure controller was incorporated in the gas line connected to the outlet to maintain a defined pressure. An online gas Thermo Fischer Trace 1300 gas chromatograph (GC) was used for product analysis. The GC was injected with 1 μl of the reactor outlet stream every 23 min. The sample was heated to either 250 °C for CoO catalysts or 450 °C for metallic Co catalysts with a 10 °C min⁻¹ ramp in a 10:20 ml min⁻¹ H₂/N₂ flow and held at that

**Table 2 Comparison of various cobalt-containing catalysts for CO/CO$_2$ conversion to C$_{2+}$ hydrocarbons.**

| Catalyst | Reactant | H$_2$/C ratio | Temperature (°C) | Pressure (bar) | Conversion (%) | C$_{2+}$ selectivity (%) | C$_{2+}$ yield (%) | C$_{2+}$ yield (ml h$^{-1}$ g$_{cat}^{-1}$) | Space time yield (l h$^{-1}$ g$_{cat}^{-1}$)[b] | Ref. |
|---|---|---|---|---|---|---|---|---|---|---|
| CoO/TiO$_2$ | CO$_2$ | 3 | 250 | 20 | 5.46 | 10.7 | 0.58 | 19.3 | 0.18[e] | This work |
| Co/TiO$_2$ | CO$_2$ | 3 | 250 | 20 | 4.12 | 4.44 | 0.18 | 6.22 | 0.14[e] | This work |
| CoO/TiO$_2$ | CO/CO$_2$ | 3 | 250 | 20 | 17.5[a] | 39.0 | 6.83 | 222 | 0.57[e] | This work |
| CoO/TiO$_2$ | CO/CO$_2$ | 3 | 250 | 20 | 18.8[a] | 13.0 | 2.43 | 79.3 | 0.61[e] | This work |
| Co-Pt-K/SiO$_2$ | CO$_2$ | 3 | 370 | 1 | 36.5 | 47.3 | 17.3 | NA[f] | NA[f] | 62 |
| Co-Na-Mo/TiO$_2$ | CO$_2$ | 3 | 200 | 80 | 13.5 | 7.69 | 1.04 | 1.54 | 0.02[e] | 63 |
| Co$_6$/MnO$_x$ | CO$_2$ | 1 | 200 | 19 | 15.1 | 53.2 | 8.03 | 5.32 | 0.01[e] | 64 |
| Co-Pt/Al$_2$O$_3$ | CO$_2$ | 1 | 220 | 10 | 6.8 | 6.9 | 0.47 | 18.8 | 0.27[c] | 65 |
| CuFe$_2$O$_4$ | CO$_2$ | 3 | 300 | 1 | 13.3 | 50.3[c] | 6.69 | 35.2 | 0.07[c] | 66 |
| K-Co/SiO$_2$ | CO$_2$ | 3 | 270 | 50 | 16.0 | 38.3 | 6.13 | 56.4 | 0.15[e] | 67 |
| 2.5K-CoCu/TiO$_2$ | CO/CO$_2$ | 3 | 250 | 20 | 13.0 | 66.4 | 8.63 | 62.2 | 0.10[e] | 68 |
| Co/TiO$_2$ | CO$_2$ | 3 | 200 | 20 | 75.3 | 60.0 | 45.2 | 118 | 0.27[e] | 69 |
| FeCo/Al$_2$O$_3$ | CO$_2$ | 3 | 320 | 20 | 49.0 | 36.9 | 18.1 | 118 | 0.32[e] | 70 |
| FeCo/NC-600 | CO$_2$ | 3 | 320 | 11 | 37.0 | 54.4 | 20.1 | 288 | 0.53[e] | 71 |
| K-FeCo/Al$_2$O$_3$ | CO$_2$ | 3 | 300 | 20 | 30.2 | 57.8 | 17.5 | 376 | 0.65[e] | 72 |
| Co/CNF | CO/CO$_2$ | 0.7 | 250 | 20 | 76.4[a] | 45.9 | 35.1 | 434 | 1.24[e] | 73 |
| CoO/TiO$_2$ | CO | 3 | 250 | 20 | 13.8 | 46.2 | 6.38 | 208 | 0.45[e] | This work |
| Co/TiO$_2$ | CO | 3 | 250 | 20 | 12.6 | 45.9 | 5.80 | 188 | 0.41[e] | This work |
| K-Fe/N-CNT | CO | 1 | 300 | 1 | 16.5 | 54.6[d] | 9.01 | 158 | 0.29[d] | 74 |
| Co-Mn | CO | 1 | 250 | 1 | 15.5 | 53.7[d] | 8.32 | 354 | 0.66[d] | 75 |

[a] Total C (CO and CO$_2$) conversion.
[b] Calculation based on reported conversion, selectivity (all products), catalyst weight, and GHSV.
[c] Molar percentage excluding CO selectivity.
[d] Molar percentage excluding CO$_2$ selectivity.
[e] Molar percentage based on all products.
[f] GHSV not indicated in this study.

temperature for 1 h. After the reduction step, the sample was cooled to 250 °C with a 10 °C min$^{-1}$ ramp. At this temperature, the gas flow was switched to 2:36:12 ml min$^{-1}$ Ar/H$_2$/CO$_2$ (GHSV = 3000 h$^{-1}$) and once the gasses were flowing, pressure was built up to 20 bar with a 1 bar min$^{-1}$ ramp. The CO$_2$ conversion and product selectivities were calculated from the following relationships:

$$X_{CO2}(\%) = \left(1 - \frac{(A_{CO2}/A_{Ar})}{(A_{CO2}^0/A_{Ar}^0)}\right) \times 100\% \qquad (1)$$

$A_{CO2}$ and $A_{Ar}$ represent the thermal conductivity detector (TCD) peak area of CO$_2$ and Ar during the reaction. $A_{CO2}^0$ and $A_{Ar}^0$ are the TCD peak areas of CO$_2$ and Ar recorded during a blank measurement. The selectivity was calculated using Eq. (2):

$$S_i(\%) = \left(\frac{(A_i \times F_i)}{(\sum_{n=1}^{\infty} A_i \times F_i)}\right) \times 100\% \qquad (2)$$

In this equation, $A_i$ corresponds to the peak area of product $i$ and $F_i$ represents the response factor of the analyte[76]. To describe the catalytic activity, CTY was used. This parameter reports the amount of CO$_2$ converted in mol per gram of cobalt per second. The parameter yield was used to describe the amounts of specific products obtained. This was reported either in mol of product per gram of cobalt per second or in (m)mol per hour per gram of catalyst.

**Operando modulated excitation infrared spectroscopy.** Operando modulated excitation ME DRIFTS experiments were conducted. The samples were sieved to a grain size of 63–250 µm and were firmly fixed in a 2 mm thick stainless-steel block which acts as a sample holder using two quartz wool plugs[77]. The sample holder was closed using a CaF$_2$ window, to allow the reflection of IR radiation, and a graphite window. The inlet of the cell was connected to two solenoid valves (Series 9, Parker), allowing fast switching needed for the modulation experiments. The outlet of the cell was connected to an online mass spectrometer (MS; Pfeiffer Vacuum Omnistar). A schematic drawing of the setup and cell is provided in Supplementary Fig. 1. The experiments were carried out with a Bruker Vertex70V Fourier Transform (FT)-IR spectrometer equipped with a liquid nitrogen cooled Mercury Cadmium Telluride detector and a Harrick Praying Mantis unit. The samples were heated to 250 °C at 10 °C min$^{-1}$ in H$_2$/N$_2$ = 1. At this temperature ten modulation periods were performed by alternating flows of CO$_2$:H$_2$ at a ratio of 1:3 (60 s) and H$_2$ (60 s). During each period of 120 s, 120 spectra were recorded at 80 KHz scanner velocity and 4 cm$^{-1}$ resolution. The ten modulation periods of 120 s each resulted in a 20 min experiment. After the modulation experiment, the sample was heated to 450 °C at 10 °C min$^{-1}$ in H$_2$/N$_2$ = 1 and was held for 1 h to reduce the metal oxide nanoparticles. Then, the sample was cooled to 250 °C at 10 °C min$^{-1}$ and the modulation experiment described above was repeated. A graphical representation of the experiment and detailed information can be found in Supplementary Fig. 2 and the accompanying text. After spectral acquisition, the sets of time-resolved data were treated by PSD[78] to obtain phase-resolved data as described in Supplementary Methods 1.6 (Supplementary Fig. 3). Phase-resolved amplitude spectra were obtained as described in Supplementary Methods 1.6.

## Data availability

The data that support the findings of this study are available within the paper and its Supplementary Information, and all data are available from the authors on reasonable request.

## Code availability

All relevant code not already in the open literature can be requested from the authors.

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

# ARTICLE

65. Dorner, R. W., Hardy, D. R., Williams, F. W., Davis, B. H. & Willauer, H. D. Influence of gas feed composition and pressure on the catalytic conversion of $CO_2$ to hydrocarbons using a traditional cobalt-based Fischer-Tropsch catalyst. *Energy Fuels* **23**, 4190–4195 (2009).

66. Choi, Y. H. et al. Carbon dioxide Fischer-Tropsch synthesis: a new path to carbon-neutral fuels. *Appl. Catal. B Environ.* **202**, 605–610 (2017).

67. Iloy, R. A. & Jalama, K. Effect of operating temperature, pressure and silica-supported cobalt catalyst in $CO_2$ hydrogenation to hydrocarbon fuel. *Catalysts* **9**, 807 (2019).

68. Shi, Z. et al. Direct conversion of $CO_2$ to long-chain hydrocarbon fuels over K-promoted CoCu/$TiO_2$ catalysts. *Catal. Today* **311**, 65–73 (2018).

69. Yao, Y., Hildebrandt, D., Glasser, D. & Liu, X. Fischer-Tropsch synthesis using $H_2$/CO/$CO_2$ syngas mixtures over a cobalt catalyst. *Ind. Eng. Chem. Res.* **49**, 11061–11066 (2010).

70. Numpilai, T. et al. Structure–activity relationships of Fe-Co/K-$Al_2O_3$ catalysts calcined at different temperatures for $CO_2$ hydrogenation to light olefins. *Appl. Catal. A Gen.* **547**, 219–229 (2017).

71. Dong, Z., Zhao, J., Tian, Y., Zhang, B. & Wu, Y. Preparation and performances of ZIF-67-derived FeCo bimetallic catalysts for $CO_2$ hydrogenation to light olefins. *Catalysts* **10**, 455 (2020).

72. Satthawong, R., Koizumi, N., Song, C. & Prasassarakich, P. Bimetallic Fe-Co catalysts for $CO_2$ hydrogenation to higher hydrocarbons. *J. CO2 Util.* **3-4**, 102–106 (2013).

73. Díaz, J. A., De La Osa, A. R., Sánchez, P., Romero, A. & Valverde, J. L. Influence of $CO_2$ co-feeding on Fischer-Tropsch fuels production over carbon nanofibers supported cobalt catalyst. *Catal. Commun.* **44**, 57–61 (2014).

74. Lu, J. et al. Promotion effects of nitrogen doping into carbon nanotubes on supported iron Fischer-Tropsch catalysts for lower olefins. *ACS Catal.* **4**, 613–621 (2014).

75. Li, Z. et al. Mechanism of the Mn promoter via CoMn spinel for morphology control: formation of $Co_2C$ nanoprisms for Fischer-Tropsch to olefins reaction. *ACS Catal.* **7**, 8023–8032 (2017).

76. Zhou, C. et al. Highly active ZnO-$ZrO_2$ aerogels integrated with H-ZSM-5 for aromatics synthesis from carbon dioxide. *ACS Catal.* **10**, 302–310 (2020).

77. Chiarello, G. L., Nachtegaal, M., Marchionni, V., Quaroni, L. & Ferri, D. Adding diffuse reflectance infrared Fourier transform spectroscopy capability to extended x-ray-absorption fine structure in a new cell to study solid catalysts in combination with a modulation approach. *Rev. Sci. Instrum.* **85**, 074102 (2014).

78. Baurecht, D. & Fringeli, U. P. Quantitative modulated excitation Fourier transform infrared spectroscopy. *Rev. Sci. Instrum.* **72**, 3782–3792 (2001).

## Acknowledgements
This work was supported by the Netherlands Research Council (NWO) in the frame of a Technology Area (TA) grant of the Innovation Fund Chemistry, together with Shell, DSM Resolve and Leiden Probe Microscopy (grant no. 731016201). Ramon Oord (Utrecht University (UU)) is gratefully acknowledged for technical support regarding the catalytic testing set-up. Mark J. Meijerink (UU) and Nienke L. Visser (UU) are acknowledged for the TEM measurements. Jelle Kranenborg (UU) is thanked for help with the ME DRIFTS experiments. Maarten Nachtegaal (PSI) is thanked for providing access to the offline laboratory of the SuperXAS beamline for the ME DRIFTS experiments. Alexander P. van Bavel (Shell International BV) is thanked for fruitful discussions on the topic. Jaap N. Louwen is gratefully acknowledged for his input on and assistance with the DFT calculations.

## Author contributions
I.C.t.H. and J.J.G.K. synthesized the catalysts and performed catalytic testing. J.J.G.K. performed the XRD measurements and Raman experiments. I.C.t.H. and M.M. performed the ME DRIFTS measurements. D.F. facilitated the ME DRIFTS measurements and F.M. facilitated the ME DRIFTS data analysis. E.B.S. performed DFT calculations. B.M.W. directed the research. I.C.t.H. wrote the manuscript with input of all co-authors.

## Competing interests
The authors declare no competing interests.
