## [Peer Review File · Nature Communications]

Title: Uncovering the Reaction Mechanism behind CoO as Active Phase for CO₂ HydrogenationREVIEWER COMMENTS

Reviewer #2 (Remarks to the Author):

The manuscript deals with the CO₂ hydrogenation performance of different cobalt based catalysts, which is an environmentally relevant and interesting topic. The conclusions drawn by the authors appear to be sound. As main results one finds (as also listed in the manuscript) (1) added evidence for CO-mediated [Co metal] and hydrogen-assisted reaction paths [CoO], (2) a high performance for CoO/TiO₂, (3) that CoO/TiO₂ may be beneficial for C₂₊ hydrocarbons and (4) that CO co-feeding increases the C₂₊ selectivity.

There is not much to be doubted about these results – they are well supported by the data. It appears that the results are relevant and may affect present views, but do not really represent new views. Also, there is not too much insight into mechanistic details. For instance, the question why Co/TiO₂-ox is such a good catalyst remains somewhat open.

Some remarks

From Fig. S4 one may have the impression that not much oxide is left over after reaction at 250° C for Co/TiO₂-ox (and others). One may wonder whether they should really be called ‘oxidic catalysts’ and what the difference between the ‘metallic’ and ‘oxidic’ catalysts after some time under reaction conditions really is.

Fig. 2 and S10 (to some extent also S12) should be better explained regarding the PSD terminology. What are the spectra in S10? Demodulated spectra for different phase shifts? Something else? In S2.5 it is said that these spectra are in the phase domain. What does this mean? Where is the phase in Fig. S10?

The caption in Fig. 2 (for 2a) says ‘Time-resolved (time-averaged) DRIFTS spectra’ Can they be time-resolved and time-averaged at the same time? What ‘time’ is actually meant? The duration of the experiment? For 2b the caption says that this panel shows ‘phase-resolved amplitude spectra’ What is this? The demodulated spectra at optimized phase or something else?

For the results to be industrially relevant, long-term studies might be required. This would show whether the conclusions do still apply after long operation times.

Reviewer #3 (Remarks to the Author):

CO₂ hydronation is an important research topic. The authors mad a good effort in material characterization in this research. The authors achieved good research results from the perspectives of catalyst activity and product selectivity under the given test conditions. However, the results are not excellent, compared to the those recently reported ones.

1. The experimental step and its graphic representation are given in Figs S1 and S2. However, Fig S2 should be better presented. Especially, the descriptive procedure should be given in detail. Otherwise, it is difficult for others to repeat the tests and check the results. For example, how was the inlet gas controlled? Were the flow rates of gases calibrated?
2. How were the activity and selectivity defined? The statement in Abstract section - "The C2+ selectivity increased to 39% (104 mmol·h⁻¹·gcat⁻¹ C2+ hydrocarbons)" is not correctly phrased.
3. What is the relationship between CTY and activity in this research?
- 4- In the most important figure (Fig 1), C1 (CH₄) has much higher selectivity. In other words, C2+ (especially C2) are not major products at all, why did authors say that "The C2+ selectivity increased to 39%"? It is confusing. Actually, 39% selectivity was achieved with Co/Al₂O₃-red for CH₄ instead of C2+, and 61% selectivity was achieved for CO, according to Fig. 1. C2 and C3 account for very small percentages of the products.
- 5- Is acidity or basicity designed for this research? Clear statements and explanation should be given.
6. How are the bonds in CO₂ (two sigma bonds and two π bonds) changed during CO₂ hydrogenation to C1-C3 products during the reaction process with the help of the catalysts?
7. What hybridization changes should we expect during the CO₂ conversion? How the catalysts used in this research help the changes?
8. Can the authors estimate the reaction order with respect to CO₂ by using the raw data collected during CO₂ conversion process?

We would genuinely like to thank the reviewers for their very valuable input, which has improved the overall quality and impact of our manuscript. In what follows, we will address the comments provided by the reviewers with a point-by-point response. Furthermore, we have highlighted in yellow or in track changes mode the alterations we made to our paper.

Reviewer #2:

We sincerely thank reviewer 2 for his/her time and very valuable feedback. Below, we will attempt to reply to the remarks provided by reviewer 2 in a step-wise manner.

Comment:

Also, there is not too much insight into mechanistic details. For instance, the question why Co/TiO₂-ox is such a good catalyst remains somewhat open.

Answer:

Reviewer 2 asked for more mechanistic details and why Co/TiO₂-ox is such a good catalyst. Co/TiO₂-ox has a more oxidic surface compared to Co/TiO₂-red and thus a lower hydrogenation capability (see also references 54-58). Because of that, the H/C ratio on the catalyst surface is lower. Consequently, less methane is formed and more olefins instead of paraffins. The metallic cobalt surface of Co/TiO₂-red has more affinity towards oxygen and can consequently directly dissociate CO_{2,ads} to CO_{ads}. On the other hand, the more oxidic surface of Co/TiO₂-ox resulted in the H-assisted mechanism as the dominant reaction mechanism. To provide more mechanistic details and to substantiate the information described above, we conducted additional experiments to determine the kinetic parameters reaction order and apparent activation energy for Co/TiO₂-ox and Co/TiO₂-red at P=20 bar and T=250°C. The apparent activation energy (E_a) for CO₂ hydrogenation was slightly lower for Co/TiO₂-ox, 113 ± 3, compared to Co/TiO₂-red, 122 ± 5 (Table R1 below, Table 1 in the main text, and sections S1.8, S2.9 in the supplementary information). This is in line with the better performance of Co/TiO₂-ox compared to Co/TiO₂-red. The higher reaction order in CO₂ of Co/TiO₂-ox (0.38 ± 0.09) compared to Co/TiO₂-red (0.15 ± 0.04) indicates that a strongly adsorbed intermediate derived from CO₂ on the Co/TiO₂-red surface, most likely CO_{ads}, hinders the reaction. Besides, the reaction orders in H₂ are almost completely opposite for the 2 samples: a positive order of 1.24 ± 0.40 for Co/TiO₂-ox versus a negative order of -1.15 ± 0.07 for Co/TiO₂-red. This particularly substantiates the hypothesis that the Co/TiO₂-ox catalyst, following the H-assisted mechanism, benefits from a higher partial pressure in H₂. On the other hand, the Co/TiO₂-red catalyst, following mainly the direct dissociation mechanism, benefits from a lower partial pressure in H₂, as H₂ may be competing with adsorbed CO, the most important intermediate in the direct dissociation mechanism.

Action taken:

We have performed additional experiments to determine the reaction orders in CO₂ and H₂, as well as experiments to determine the apparent activation energies (Table 1 in the main text). Besides, we have performed density functional theory (DFT) calculations to highlight the difference between metallic cobalt and cobalt oxide (Supplementary Figure 11). The results have been added to the main text and to the supplementary information (SI).

Table R1. Kinetic parameters for Co/TiO₂.

Catalyst	E _a (kJ/mol) ^a	R ²	CO ₂ reaction order ^b	H ₂ reaction order
Co/TiO ₂ -ox	113 ± 3	0.98	0.38 ± 0.09	1.24 ± 0.40
Co/TiO ₂ -red	122 ± 5	0.99	0.15 ± 0.04	-1.15 ± 0.07

- ^a E_a was calculated with 6 data points between 200 and 280°C and the R² values are reported in the 3rd column.
- ^b CO₂ and H₂ reaction orders were determined at 250°C.

Comment:

From Fig. S4 one may have the impression that not much oxide is left over after reaction at 250°C for Co/TiO₂-ox (and others). One may wonder whether they should really be called ‘oxidic catalysts’ and what the difference between the ‘metallic’ and ‘oxidic’ catalysts after some time under reaction conditions really is.

Answer:

In Figure S4 (now Supplementary Figure 5), reviewer 2 pointed out that the amount of cobalt oxide (deducted from the Raman spectra) during CO₂ hydrogenation at 250°C decreased compared to the fresh catalyst and that “one may wonder what the difference between the ‘metallic’ and ‘oxidic’ catalysts after some time under reaction conditions really is.”. The decreased amount of cobalt oxide is particularly visible in the Co/TiO₂-ox and Co/CeO₂-ox *operando* Raman spectra. We would like to emphasize that the fresh catalyst contains Co₃O₄, which transforms into CoO during the activation at 250°C in H₂. In terms of the spectral fingerprint, this means that the characteristic Co-O vibrations (*e.g.*, ~690 cm⁻¹) become broader, less intense, and shift to a slightly higher frequency. Hence, compared to the fresh catalyst, the catalyst under reaction conditions will always have a slightly different Raman spectrum. We would like to refer here to Supplementary Figure 9, where we have performed H₂-temperature programmed reduction (TPR) to verify the oxidation state under H₂ atmosphere at increasing temperatures. For all catalysts, the first peak signifying the Co₃O₄ to CoO transition appears between 250°C and 290°C. Though, due to the invisibility of metallic Co in the Raman spectra (metallic Co is Raman inactive), we cannot unambiguously conclude that no metallic Co is present under CO₂ hydrogenation reaction conditions at all. However, based on the Raman and H₂-TPR results, we do believe that when the catalyst is heated to 250°C only, CoO will be the dominant oxidation state present under CO₂ hydrogenation reaction conditions, which is why we named this set of catalysts with the suffix “-ox”. To support this hypothesis, we additionally performed X-ray diffraction (XRD) on Co/TiO₂-ox and Co/TiO₂-red after 150 h of catalytic testing. As can be seen in Supplementary Figure 23, Co/TiO₂-ox contained CoO and Co/TiO₂-red contained metallic cobalt (face-centered cubic (FCC)).

Action taken:

We have additionally performed X-ray diffraction (XRD) on our spent -ox and -red samples for the Co/TiO₂ catalyst. The results have been added to the SI (Supplementary Figure S23) and a referring sentence has been added to the main text.

Comment: *Fig. 2 and S10 (to some extent also S12) should be better explained regarding the PSD terminology. What are the spectra in S10? Demodulated spectra for different phase shifts? Something else? In S2.5 it is said that these spectra are in the phase domain. What does this mean? Where is the phase in Fig. S10?*

The caption in Fig. 2 (for 2a) says ‘Time-resolved (time-averaged) DRIFTS spectra’ Can they be time-resolved and time-averaged at the same time? What ‘time’ is actually meant? The duration of the experiment? For 2b the caption says that this panel shows ‘phase-resolved amplitude spectra’ What is this? The demodulated spectra at optimized phase or something else?

Answer:

We thank the reviewer, we realized that our terminology could have been clearer. We answer the two comments in the following. ‘Time-resolved (time averaged) DRIFTS spectra’ in the caption of Figure 2 was misleading. Averaging means that each i^{th} spectrum in a modulation period consisting of x_i spectra ($x = 120$) is averaged along all modulation periods (10) to provide 120 average spectra (Supplementary Figure 2). The experiments consisted of 10 modulations ($\text{CO}_2 + \text{H}_2$ vs. H_2) of 120 s each, which results in a duration of 20 min. Figure 2a shows these averaged spectra of a particular sample. The averaged spectra were treated by phase-sensitive detection (PSD) to obtain the phase-resolved spectra (Supplementary Figure 3). In this work, the phase-resolved spectra were further elaborated by determining the absolute maximum value at each wavenumber to avoid the co-existence of positive and negative signals. The resulting phase-resolved amplitude spectra are displayed in Figure 2b.

Action taken:

More detailed information was added to the SI to improve clarity. The caption of Figure 2: (time-averaged) was omitted and a sentence was added to direct the reader to the SI for further details on the experimental methodology. The caption of Supplementary Figure 3: the text was improved to hopefully make it clearer how the data shown in the main text and in the SI have been obtained.

Comment:

For the results to be industrially relevant, long-term studies might be required. This would show whether the conclusions do still apply after long operation times.

Answer:

Reviewer 2 pointed out that long-term catalytic testing studies should be performed in order to verify industrial relevance. To answer this question, we have performed additional experiments where we tested Co/TiO₂-ox for 150 h in total at T=250°C and P=20 bar. First for 50 h under CO₂ hydrogenation conditions while co-feeding CO (CO₂/CO=2) and then for 100 h under CO₂ hydrogenation conditions (H₂/CO₂=3). For the 50 h of co-feeding, the total carbon conversion started at ~18% and stabilized after about 10 h to ~16%, while the C₂₊ selectivity started at ~40% and stabilized at ~35%. For the following 100 h of CO₂ conversion only, the conversion started at ~7.0% and remained ~4.5% after 100 h, while the C₂₊ selectivity increased from ~10% in the first few h to ~20% after 100 h, indicating that the activity loss over time was mostly related to a decrease in methane production. The resulting figures have been added to Figure 6 (c, d) in the main text. Besides, we have performed additional experiments during which we varied the reactant concentration and temperature in order to determine a few additional key kinetic parameters (see the SI section S2.9 and Table R1 above for details).

Action taken:

We have performed additional experiments where we tested Co/TiO₂-ox for 150 h in total at T=250°C and P=20 bar. First for 50 h under CO₂ hydrogenation conditions while co-feeding CO (CO₂/CO=2) and then for 100 h under CO₂ hydrogenation conditions (H₂/CO₂=3).

Reviewer #3:

We sincerely thank reviewer 3 for his/her time and very valuable feedback. Below, we will attempt to reply to the remarks provided by reviewer 3 in a step-wise manner.

Comment:

1. The experimental step and its graphic representation are given in Figs. S1 and S2. However, Fig. S2 should be better presented. Especially, the descriptive procedure should be given in detail.

Otherwise, it is difficult for others to repeat the tests and check the results. For example, how was the inlet gas controlled? Were the flow rates of gases calibrated?

Answer:

We realized that we did not appropriately describe the data analysis and used unclear terminology. The gas inlets were controlled with fast switching (solenoid) valves and the flow rates were controlled using (calibrated) mass flow controllers (MFCs).

Action taken:

More detailed information was added to the experimental section of the main text and to the supplementary information (SI) (Supplementary section S1.6 and Supplementary Figures 2-3) to improve clarity. The caption of Figure 2: (time-averaged) was omitted and a sentence was added to direct the reader to SI for further details on the experimental methodology. The caption of Supplementary Figure 3: the text was improved to hopefully make it clearer how the data shown in the main text and in the SI have been obtained.

Comments:

- 2. How were the activity and selectivity defined? The statement in Abstract section - “The C₂₊ selectivity increased to 39% (104 mmol·h⁻¹·g_{cat}⁻¹ C₂₊ hydrocarbons)” is not correctly phrased.*
- 3. What is the relationship between CTY and activity in this research?*

Answer:

Reviewer 3 inquired about the relationship between CTY and activity in our study. Details on how the activity and selectivity were defined and determined from the gas chromatography data can be found in the Methods section under “High pressure catalytic testing”. Generally, catalytic activity is reported as percentage of reactant (CO₂) converted or as metal(cobalt)-time-yield (CTY). CTY is defined as mol CO₂ converted per gram of cobalt per second. Alternatively, activity can be reported as the absolute percentage of CO₂ that is converted by the catalyst (often referred to as “activity”). In this work, we report both the measures. See for example Figure 1i (CTY), Table 2, and Supplementary Tables 5 and 9 (%). The selectivity is defined as % of product x in the total products. Besides, we also reported specific product yields per gram of catalyst per hour, as this information would be relevant to make an assessment of *e.g.*, cost effectiveness. For example, in the abstract we reported “The C₂₊ selectivity increased to 39% (104 mmol·h⁻¹·g_{cat}⁻¹ C₂₊ hydrocarbons)”, which appeared somewhat confusing to the reviewer. We would like to apologize for that and we have added a few clarifying/rephrased sentences in the Methods section.

Action taken:

We have added a few clarifying/rephrased sentences in the Methods section on how cobalt-time-yield (CTY) and specific product yields were determined.

Comment:

4- In the most important figure (Fig 1), C1 (CH₄) has much higher selectivity. In other words, C₂₊ (especially C₂) are not major products at all, why did authors say that “The C₂₊ selectivity increased to 39%”? It is confusing. Actually, 39% selectivity was achieved with Co/Al₂O₃-red for CH₄ instead of C₂₊, and 61% selectivity was achieved for CO, according to Fig. 1. C₂ and C₃ account for very small percentages of the products.

Answer:

CH₄ was almost always the main product during CO₂ hydrogenation, apart from Co/Al₂O₃-red, which produced 61% CO. The 39% C₂₊ selectivity (as described in the abstract) was achieved with Co/TiO₂-ox upon co-feeding CO to the gas stream during CO₂ hydrogenation (CO₂/CO=2). For CO₂ hydrogenation only, the C₂₊ selectivity was 11% for that catalyst.

Action taken:

We have rewritten this to hopefully make it clearer. Clarifying sentences have been added to the Methods section on how specific product yields were determined.

Comment:

5- *Is acidity or basicity designed for this research? Clear statements and explanation should be given.*

Answer:

Reviewer 3 asked how the acidity and basicity are designed in our study. We are assuming that reviewer 3 means acidity or basicity of the support materials. These properties were previously not experimentally determined in our study. However, during the revisions we have performed CO₂-temperature programmed desorption on the support materials and defined the basicity as the amount of CO₂ desorbed per unit surface area. The results indicate that the basicity increases in the order SiO₂<Al₂O₃< CeO₂<TiO₂. We have added a short statement to clarify this in the main text and a more elaborate explanation in the SI. We used the general knowledge about the higher basicity of *e.g.*, TiO₂ and CeO₂ support (reducible) to describe and explain the formation of large amounts of carbonates, bicarbonates, and formates on the catalyst surface during the CO₂ hydrogenation reaction in a qualitative manner.

Action taken:

We have performed CO₂-temperature programmed desorption on the support materials to experimentally determine the basicity.

Comment:

6. *How are the bonds in CO₂ (two sigma bonds and two π bonds) changed during CO₂ hydrogenation to C1-C3 products during the reaction process with the help of the catalysts?*

Answer:

Reviewer 3 asked how the bonds in CO₂ change during CO₂ hydrogenation to C₁-C₃ hydrocarbon products and what the role of the catalysts is herein. The 2 π bonds and 2 σ bonds in CO₂ are changed during the catalytic conversion into C₁-C₃ products. When CO₂ molecules interact with metal (oxide) surfaces, electrons are donated from the metal d-orbitals to the CO₂ π* orbitals. This weakens and eventually breaks the C-O bond(s). In case of the direct dissociation mechanism, which is dominant for the metallic cobalt catalysts, the resulting O_{ads} is hydrogenated and desorbs as H₂O, while the C_{ads} is hydrogenated stepwise with H_{ads}. If the C_{ads} is fully hydrogenated and desorbs from the surface, C₁ products (methane) are formed. Alternatively, multiple adsorbed CH₂ units can for example combine to form C₂₊ products. In case of the H-assisted mechanism, which is dominant for the CoO catalysts, hydrogen assists in weakening the C-O bonds *via e.g.*, HCO intermediates. This is necessary, given that the CoO surface is less electronegative compared to the metallic Co surface and can likely not donate enough electrons to the CO₂ π* orbitals to break the bonds without the help of hydrogen. The oxygen moieties are hydrogenated and desorb as H₂O. The resulting adsorbed CH or CH₂ units can either be fully hydrogenated to form methane or undergo C-C coupling to form C₂₊ products. C-C coupling may be more likely to occur on the

CoO surface than on the metallic Co surface, due to the lower hydrogenation capability of CoO (see also references 54-58). Because of that, the H/C ratio on the CoO-containing catalyst surface is lower. Consequently, less methane is formed and more olefins instead of paraffins. To obtain a more theoretical understanding, we performed density functional theory (DFT) calculations and compared the adsorption energy, O-C-O bond angle deformation, and C-O bond elongation for adsorbed CO₂ on the CoO and metallic Co surface.

Action taken:

We have made a more elaborate scheme for the elementary reaction steps in the CO₂ hydrogenation reaction and added this scheme to the main text as Figure 5. Besides, we have performed DFT calculations to obtain a more theoretical understanding (Supplementary Figure 11).

Comment:

7. What hybridization changes should we expect during the CO₂ conversion? How the catalysts used in this research help the changes?

Answer:

Reviewer 3 asked what hybridization changes we can expect during CO₂ conversion and how the catalysts aid these changes. Initially, the gaseous CO₂ molecules are sp hybridized. During the conversion, one can expect that the hybridization changes to a different state multiple times, as bonds are broken and new bonds are formed. Upon adsorption of CO₂ on the catalyst surface the vibrational frequency changes from ~2360 cm⁻¹ to ~2344 cm⁻¹ (see also Supplementary Table 7), indicating bond weakening. The hybridization remains sp when the CO₂ is adsorbed in linear configuration. However, in a more trigonal configuration (strong adsorption, see Figure 2h) hybrid orbitals are formed and the CO_{2,ads} hybridization changes to sp². For the direct dissociation mechanism, dominant for the metallic Co catalysts, CO_{ads} is formed when electrons from the catalyst's metal d-orbitals move into the CO₂ antibonding orbitals and one of the C-O bonds is broken. CO_{ads} is sp hybridized when linearly adsorbed. The C-O bond of CO_{ads} is again broken by electrons from the catalyst's metal d-orbitals moving into the CO antibonding orbitals. To form reaction products, the resulting C_{ads} is then hydrogenated stepwise to CH_{ads} (sp), CH_{2,ads} (sp²), and potentially CH_{3,ads} (sp³) and CH₄ (gas, sp³). Other products may include paraffins (sp³) and olefins (sp²). For the H-assisted mechanism, adsorbed HCO₂²⁻ is formed as an intermediate. This has a tetragonal configuration and is sp³ hybridized. Adsorbed HCO is formed upon further hydrogenation and has a trigonal configuration and is sp² hybridized. The stepwise hydrogenation into products goes again *via* CH_{ads} (sp), CH_{2,ads} (sp²), and potentially CH_{3,ads} (sp³) and CH₄ (gas, sp³). Other products may include paraffins (sp³) and olefins (sp²). The cobalt-based catalysts aid this conversion because they donate electrons from the metal d-orbitals to the CO₂ π* orbitals, which eventually breaks the C-O bonds. To obtain a more theoretical understanding, we performed density functional theory (DFT) calculations and compared the adsorption energy, O-C-O bond angle deformation, and C-O bond elongation for adsorbed CO₂ on the CoO and metallic Co surface.

Action taken:

We have performed DFT calculations to illustrate how the hybridization of CO₂ changes upon adsorption on the CoO and metallic Co surface. A few lines of text have been added to the main text and more elaborate information can be found in the SI (Supplementary Figure 11).

Comment:

8. Can the authors estimate the reaction order with respect to CO₂ by using the raw data collected during CO₂ conversion process?

Answer:

Reviewer 3 asked if we could estimate the reaction order in CO₂ based on catalytic data. We thank reviewer 3 for this question and certainly followed up on this. To determine the reaction order with respect to CO₂, we have conducted additional experiments with the Co-TiO₂-ox and Co-TiO₂-red catalysts in which we varied the CO₂ concentration during CO₂ hydrogenation reaction while keeping the H₂ concentration constant. The results indicate a reaction order in CO₂ of 0.38±0.09 for Co-TiO₂-ox and 0.15± 0.04 for Co-TiO₂-red at T=250°C and P=20 bar (see Table R1 below, Table 1 in the main text and the Supplementary sections S1.8 and S2.9). The higher reaction order in CO₂ of Co/TiO₂-ox (0.38 ± 0.09) compared to Co/TiO₂-red (0.15 ± 0.04) indicates that a strongly adsorbed intermediate derived from CO₂ on the Co/TiO₂-red surface, most likely CO_{ads}, hinders the reaction. To determine the reaction order in H₂, we also varied the H₂ concentration while keeping the CO₂ concentration constant. We found that the reaction orders in H₂ are almost completely opposite for the 2 samples: a positive order of 1.24 ± 0.40 for Co/TiO₂-ox versus a negative order of -1.15 ± 0.07 for Co/TiO₂-red. This particularly substantiates the hypothesis that the Co/TiO₂-ox catalyst, following the H-assisted mechanism, benefits from a higher partial pressure in H₂. On the other hand, the Co/TiO₂-red catalyst, following mainly the direct dissociation mechanism, benefits from a lower partial pressure in H₂, as H₂ may be competing with adsorbed CO, the most important intermediate in the direct dissociation mechanism. We had already determined the activation energies for all catalysts P=1 bar (Supplementary Figure 8). But with additional experiments, we also determined other kinetic parameters for the Co-TiO₂ catalysts at P=20 bar, such as the apparent activation energy E_a, by varying the reaction temperature (see Table R1 below, Table 1 in the main text and the Supplementary sections S1.8 and S2.9). Besides, we performed long-term stability testing for 150 hours in total With Co/TiO₂-ox at T=250°C and P=20 bar. First for 50 h under CO₂ hydrogenation conditions while co-feeding CO (CO₂/CO=2) and then for 100 h under CO₂ hydrogenation conditions (H₂/CO₂=3). For the 50 h of co-feeding, the total carbon conversion started at ~18% and stabilized after about 10 h to ~16%, while the C₂₊ selectivity started at ~40% and stabilized at ~35%. For the following 100 h of CO₂ conversion only, the conversion started at ~7.0% and remained ~4.5% after 100 h, while the C₂₊ selectivity increased from ~10% in the first few h to ~20% after 100 h, indicating that the activity loss over time was mostly related to a decrease in methane production.

Table R2. Kinetic parameters for Co/TiO₂.

Catalyst	E _a (kJ/mol) ^a	R ²	CO ₂ reaction order ^b	H ₂ reaction order
Co/TiO ₂ -ox	113 ± 3	0.98	0.38 ± 0.09	1.24 ± 0.40
Co/TiO ₂ -red	122 ± 5	0.99	0.15 ± 0.04	-1.15 ± 0.07

^a E_a was calculated with 6 data points between 200 and 280°C and the R² values are reported in the 3rd column.

^b CO₂ and H₂ reaction orders were determined at 250°C.

Action taken:

We have performed additional experiments to determine the reaction orders in CO₂ and H₂, as well as experiments to determine the apparent activation energies. Besides, we determined the long-term stability over 150 h time-on-stream for Co/TiO₂-ox, the best performing catalyst in our study.

REVIEWERS' COMMENTS

Reviewer #2 (Remarks to the Author):

The detail comments on the first submission have been sufficiently treated. The results are well-supported by the data and the conclusions are probably correct. As written before, the results are relevant and may affect present views, but do not really represent new views. Nevertheless, the weight might be sufficient for publication.

Reviewer #3 (Remarks to the Author):

The revision work was done.

REVIEWERS' COMMENTS

Reviewer #2 (Remarks to the Author):

The detail comments on the first submission have been sufficiently treated. The results are well-supported by the data and the conclusions are probably correct. As written before, the results are relevant and may affect present views, but do not really represent new views. Nevertheless, the weight might be sufficient for publication.

Answer:

We would like to thank reviewer #2 for their time and effort. The input given by reviewer #2 motivated us to perform long-term stability studies and additional X-ray diffraction measurements. The feedback provided by reviewer #2 has certainly improved the quality of the manuscript.

Reviewer #3 (Remarks to the Author):

The revision work was done.

Answer:

We would like to thank reviewer #3 for their time and effort. The input given by reviewer #3 motivated us to perform density functional theory calculations, kinetic studies, and basicity measurements. The feedback provided by reviewer #3 has certainly improved the quality of the manuscript.